# Three-dimensional holographic imaging of incoherent objects through scattering media

YoonSeok Baek ✉, Hilton B. de Aguiar & Sylvain Gigan

Three-dimensional (3D) high-resolution imaging is essential in microscopy, yet light scattering poses significant challenges in achieving it. Here, we present an approach to holographic imaging of spatially incoherent objects through scattering media, utilizing a virtual medium that replicates the scattering effects of the actual medium. This medium is constructed by retrieving mutually incoherent fields from the object and exploiting the spatial correlations between them. By numerically propagating the incoherent fields through the virtual medium, we non-invasively compensate for scattering, achieving accurate 3D reconstructions of hidden objects. Experimental validation with fluorescent and synthetic incoherent objects confirms the effectiveness of this approach, opening new possibilities for advanced 3D high-resolution microscopy in scattering environments.

Visualizing the 3D structures of materials and biological specimens is fundamental to various scientific fields. To capture these structures at the microscopic scale, conventional microscopy techniques achieve 3D imaging by controlling excitation beams[1], employing engineered point spread functions[2–4], or utilizing holographic approaches[5–8]. However, these methods, which depend on precisely controlled excitation and detection schemes, face significant limitations in highly scattering environments where light scattering generates random speckle patterns, obscuring spatial information.

Over the past decade, several methods have been developed for non-invasive imaging through scattering media[9,10]. For imaging objects that reflect or emit spatially incoherent light, such as fluorescent sources, referred to as incoherent objects, wavefront shaping techniques achieve focused excitation through scattering media by exploiting nonlinear fluorescence signals[11–14] or optimizing specific metrics[15–18]. Alternatively, computational imaging using scattered light[19–24] has been demonstrated by exploiting speckle correlations[25]. Despite their differences, both approaches are generally restricted to two-dimensional (2D) imaging of small areas, due to the limited range of focus scanning and speckle correlations. More recently, efforts have been made towards 3D incoherent imaging through scattering media[26–31], but they rely on simplifying assumptions such as treating fluorescence as fully coherent field or requiring the object to be distant from the scattering plane.

Here, we introduce a method for holographic imaging of spatially incoherent objects obscured by scattering media. Our approach begins by retrieving multiple incoherent fields through spatial modulation of scattered light, as recently demonstrated in ref. 32. Leveraging the inherent correlation within the scattered fields, we extract wavefront distortions at a specific internal plane to build a virtual medium that replicates the scattering effects of the actual medium. By numerically propagating the scattered fields through this virtual medium, our method effectively compensates for the scattering, reconstructing 3D images of the hidden objects (Fig. 1). This field-based approach extends non-invasive imaging capabilities beyond the conventional limits of intensity-based methods, extending both lateral and axial imaging ranges. We experimentally demonstrate the effectiveness of this method by imaging incoherent objects, composed of fluorescent and synthetic sources, through scattering layers.

## Results

Our method constructs a virtual medium that replicates the scattering behavior of the actual physical medium by leveraging the correlation between scattered fields. To obtain the fields of spatially incoherent light, we employed a phase retrieval method that we recently introduced in ref. 32. This method spatially modulates the incoherent light using random phase masks. Then the resulting speckle pattern is analyzed through an iterative algorithm to retrieve mutually

Laboratoire Kastler Brossel, ENS-Université PSL, CNRS, Sorbonne Université, Collège de France, Paris, France. ✉e-mail: yoonseok.baek@lkb.ens.fr

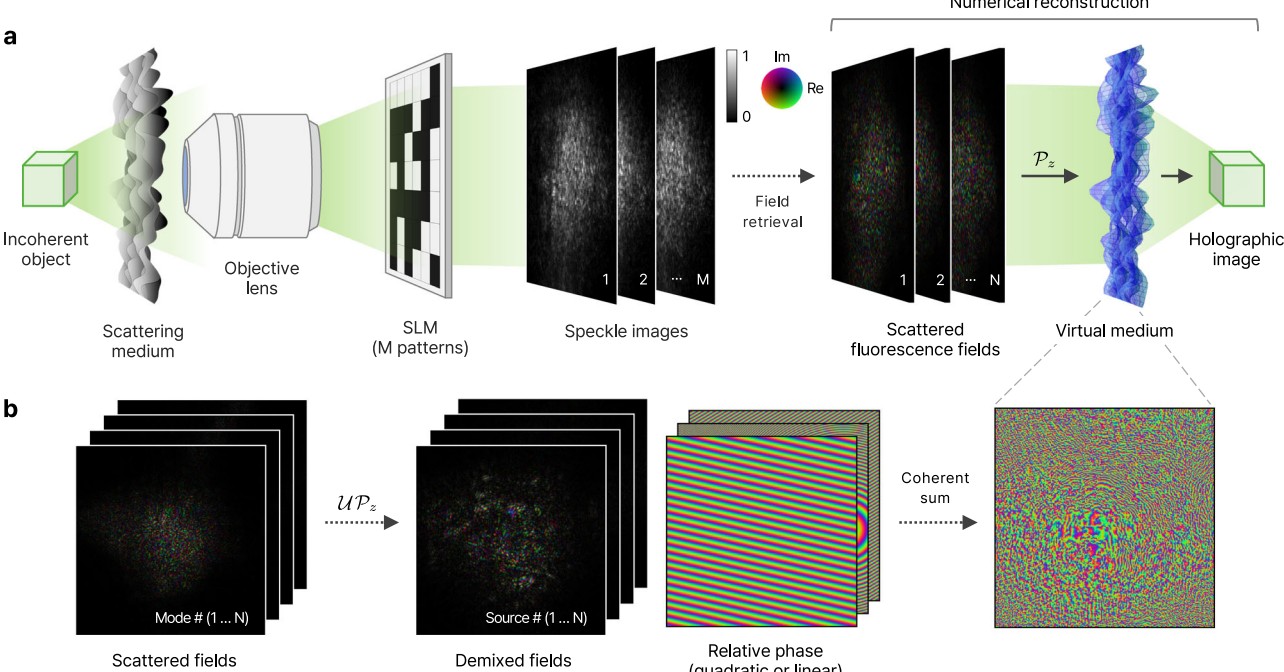

**Fig. 1 | Holographic reconstruction of incoherent objects using virtual medium. a** Experimental setup and image reconstruction. The scattering medium scatters the incoherent light emitted by an object. The scattered light is then spatially modulated by random patterns on a spatial light modulator (SLM). The resulting low-contrast speckle images are used to retrieve scattered fields using the phase retrieval algorithm. By numerically propagating the fields through a virtual scattering medium, the image of the object is obtained. **b** Reconstruction of the virtual medium. The retrieved fields are numerically propagated ($\mathcal{P}_z$) and unitary transformed ($\mathcal{U}$) into the scattered fields for individual sources. After correcting their relative phase (quadratic or linear), the scattered fields are coherently summed to construct the virtual medium.

incoherent fields of independent incoherent sources (see Methods for details on the field retrieval and analysis).

The underlying correlation between these scattered fields generally manifests as shifts and tilts in their wavefronts for neighboring, laterally shifted sources[33]. This correlation can be expressed as

$$C(\Delta \mathbf{r}, \Delta \mathbf{k}) = \int E_n(\mathbf{r} - \Delta \mathbf{r}) E_m^*(\mathbf{r}) e^{i\Delta \mathbf{k} \cdot \mathbf{r}} d^2 \mathbf{r}. \qquad (1)$$

Here $\mathbf{r} = (x, y)$ and $E_n(\mathbf{r})$ is the scattered field from an $n$-th source, corresponding to a column of an outgoing transmission matrix. In forward-scattering scenarios, this correlation reaches its maximum when the shift $\Delta \mathbf{r}$ and tilt $\Delta \mathbf{k}$ are optimally adjusted[33]. At a specific internal plane, these adjustments are equivalent to a wavefront tilt without any lateral shift:

$$\max|C| = \left| \int \mathcal{E}_n(\mathbf{r}) \mathcal{E}_m^*(\mathbf{r}) e^{i\Delta \mathbf{k}_{\text{opt}} \cdot \mathbf{r}} d^2 \mathbf{r} \right|, \qquad (2)$$

where $\mathcal{E}_n(\mathbf{r}) = \mathcal{P}_{\hat{z}}[E_n(\mathbf{r})]$, $\mathcal{P}_{\hat{z}}$ is the propagation to the plane where the tilt correlation is maximized, and $\Delta \mathbf{k}_{\text{opt}}$ is the optimal wavefront tilt. This expression implies that, after accounting for the wavefront tilt, scattered fields retain a consistent wavefront distortion caused by scattering. We can represent this distortion as $S_o(\mathbf{r})$, at the internal plane, such that $\mathcal{E}_n(\mathbf{r}) \approx S_o(\mathbf{r})h_n(\mathbf{r})$, where $h_n(\mathbf{r})$ represents the field emitted by the source under free-space propagation (See Supplementary for details and Methods for locating the correlation plane). Although the argument above primarily applies to laterally shifted sources, it can be extended to three-dimensional objects. Specifically, if the axial extent of the incoherent object is not significantly larger than the scattering mean free path, scattering within the object's axial size can be neglected. Under this assumption, the fields retain the form

$S_o(\mathbf{r})h_n(\mathbf{r})$, with $h_n$ determined by the 3D positions of the sources. We can then define a virtual scattering layer $S(\mathbf{r})$ to represent the wavefront distortion as

$$\arg[S(\mathbf{r})] = \arg\left[\sum_n \mathcal{E}_n(\mathbf{r}) e^{i\Phi_{nm}(\mathbf{r})}\right], \qquad (3)$$

where $\Phi_{nm}(\mathbf{r})$ represents the quadratic phase term, given by the phase of $h_n^*(\mathbf{r})h_m(\mathbf{r}) \approx \mathcal{E}_n^*(\mathbf{r})\mathcal{E}_m(\mathbf{r})$, and $m$ is an arbitrary index. For 2D objects, $\Phi_{nm}(\mathbf{r})$ reduces to the optimal wavefront tilt in Eq. (2). Equation (3) describes the wavefront distortion that is common to all scattered fields relative to $h_m^*(\mathbf{r})$, representing the phase of $S_o(\mathbf{r})h_m(\mathbf{r})$ rather than $S_o(\mathbf{r})$ alone. Nonetheless, by propagating the fields through this layer, the scattering effects can be effectively compensated, resulting in a 3D image of the incoherent object. The corresponding holographic image is obtained as:

$$I(\mathbf{r}, z) = \sum_n |\mathcal{P}_z\{\mathcal{F}[\mathcal{E}_n^* S](\mathbf{r})\}|^2, \qquad (4)$$

where $\mathcal{F}$ is the Fourier transformation (see Supplementary for further details). In practice, accurately determining the relative phase $\Phi_{nm}(\mathbf{r})$ could be challenging when there is limited spatial overlap between the fields $\mathcal{E}_n(\mathbf{r})$ and $\mathcal{E}_m(\mathbf{r})$. To address this, we initially constructed $S(\mathbf{r})$ using a pair of fields with high correlation, then incrementally updated $S(\mathbf{r})$ by incorporating additional fields that show strong correlation (see Methods).

To experimentally demonstrate the method, we used fluorescent beads placed approximately $100\ \mu m$ behind a highly scattering layer (see Methods for experimental setup). The scattered fluorescence produced an intensity sum of speckle patterns (Fig. 2a, e), and the speckle autocorrelation failed to reveal information about the object

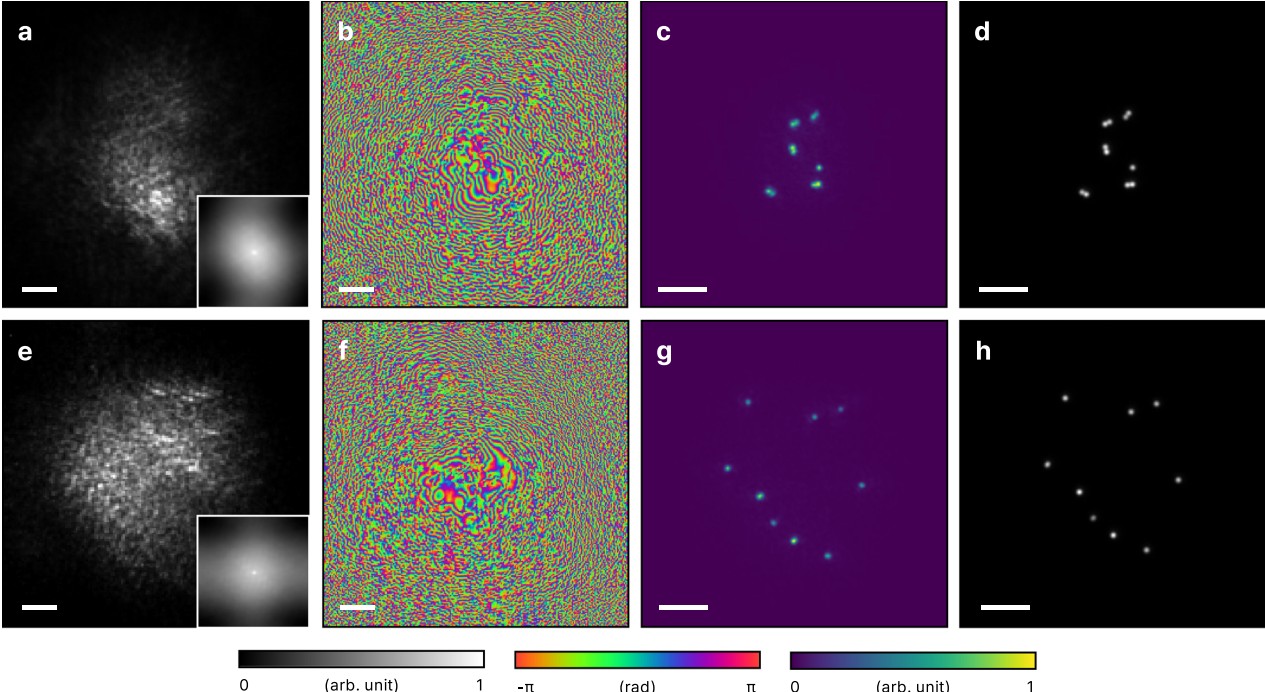

**Fig. 2 | Imaging 2D fluorescent objects through separate scattering layers.**
**a**, **e** Fluorescence speckle measured at the camera. (Inset) Background-subtracted speckle auto-correlation. **b**, **f** Virtual scattering layers constructed using the scattered fluorescence fields. **c**, **g** Image obtained by numerically propagating the fields through the virtual scattering layers. **d**, **h** Ground truth fluorescence images measured from the side without the scattering layers. Scale bars represent 10 μm.

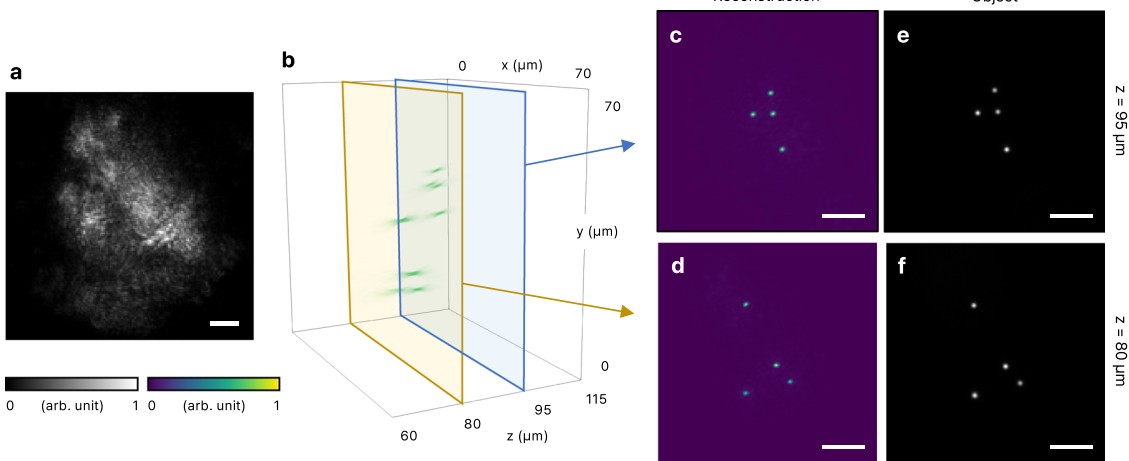

**Fig. 3 | Imaging a 3D fluorescent object through a scattering layer.**
**a** Fluorescence speckle measured at the camera. **b** 3D rendering of the numerical propagation. **c**, **d** Image reconstructed at 80 and 95 μm away from the virtual layer. **e**, **f** ground truth image obtained from the side without the scattering layer. Scale bars represent 10 μm.

(Fig. 2a, e insets). This is because, under microscopic imaging conditions with the object near the scattering layer, scattering cannot be approximated as a convolution of the object with a single speckle pattern[34]. Despite this limitation, our method successfully retrieved incoherent scattered fields and constructed the virtual scattering layer (Fig. 2b, f). Each virtual layer was numerically positioned at the plane where the field correlation was highest. Finally, images of the hidden fluorescent objects were reconstructed by numerically propagating the fields through the virtual layer and summing their intensities based on Eq. (4) (Fig. 2c, g). The reconstructed images are in strong agreement with the fluorescence images taken without the scattering layer (Fig. 2d, h). The peak signal-to-noise ratio (PSNR) and structural similarity index measure (SSIM) between the reconstruction and ground truth images are 35.5 dB and 0.96 for Fig. 2c, d, 38.1 dB and 0.93 for Fig. 2g, h, respectively. These results confirm the effectiveness of our method.

An important advantage of our holographic approach is its capability for 3D imaging through scattering media. To demonstrate this, we placed fluorescent beads at two different depths behind the scattering layer (see Methods). As in previous experiments, the scattered fluorescence produced low-contrast speckle patterns on the camera (Fig. 3a). We then constructed a virtual scattering layer and numerically propagated the scattered fields through it (see Methods). Figure 3b shows the resulting 3D intensity distribution from this propagation, where z represents the distance from the physical scattering layer. The magnification and z-coordinate of the image were adjusted for

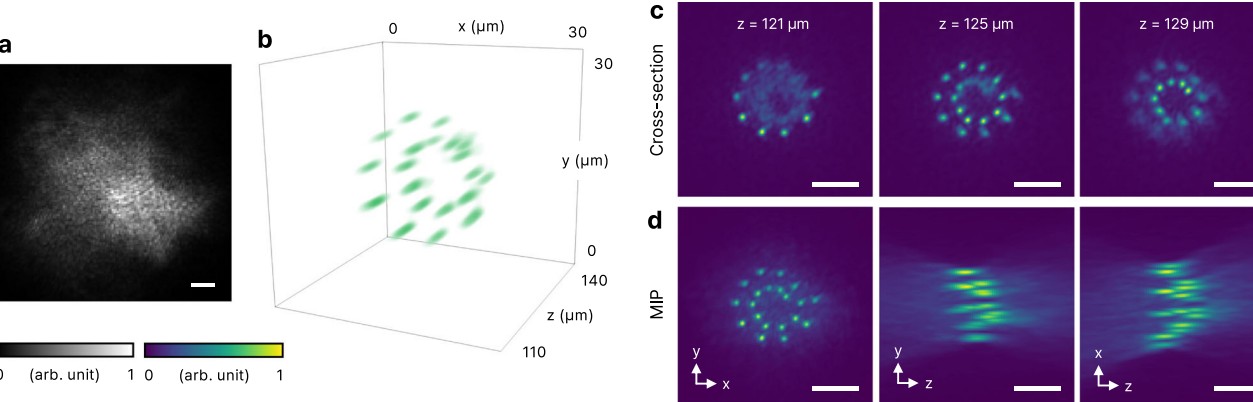

**Fig. 4 | Imaging a complex synthetic incoherent object through a scattering layer. a** Experimental speckle image of the synthetic incoherent object arranged in a spiral. **b** Rendered 3D reconstruction of the object. **c** Cross-sectional view of the reconstruction. **d** Maximum intensity projection (MIP) images of the reconstruction. Scale bars represent 10 μm.

accurate representation (see Supplementary). The result clearly reveals the 3D distribution of fluorescent sources, with images at the two planes highly consistent with those obtained without the scattering layer (Fig. 3c–f). The PSNR and SSIM were 40.1 dB and 0.91 for Fig. 3c, e, and 39.9 dB and 0.95 for Fig. 3d, f, respectively.

To demonstrate the capability of our method in handling more complex scenarios, we tested its performance with sources distributed across multiple axial planes using a synthetic 3D incoherent object. The incoherent data were generated by projecting diffraction-limited laser foci, corresponding to point sources, in a 3D volume using a digital micromirror device (see Methods), which served as ground truth. Speckle images from all point sources were measured and summed to simulate the final incoherent image (Fig. 4a), which was then processed by the algorithm as if it originated from a single incoherent object. In this experiment, 20 diffraction-limited foci were arranged along a spiral trajectory in cylindrical coordinates, described by the parametric equation $(\rho_n, \varphi_n) = (\rho_1 - \rho_2 \frac{n-1}{20}, 4\pi \frac{n-1}{20})$, where $n$ is the index of each source, $\rho_1 = 10$ μm and $\rho_2 = 6.7$ μm. The axial position of each source was increased by 0.5 μm, resulting in a total span of 10 μm. The reconstructed 3D image successfully captured the spatial distribution of the incoherent object (Fig. 4b). The 3D positions of the sources were clearly visualized in cross-sectional views (Fig. 4c) and maximum intensity projections (MIPs, Fig. 4d). Our reconstruction is analogous to digital holography once the virtual scattering medium is identified. The resolution is set by the spatial frequency content of the light, not by the reconstruction method itself, since our approach numerically propagates the retrieved fields. For our experimental setup with an effective NA of 0.35, the corresponding resolution is 0.96 μm. The full width at half maximum of a source in Fig. 4c is 1.04 μm, showing close agreement. To demonstrate positional accuracy, we overlaid the expected positions of the point sources on the MIP images (Supplementary Fig. S1). To account for the difference in viewing angle between the illumination and detection setups, we applied a rotation correction for optimal alignment. The positional error in all directions is comparable to or below the diffraction limit. To further demonstrate the versatility of the method, we conducted numerical simulations with incoherent objects shaped as three perpendicular lines and cube edges (Supplementary Fig. S2). These objects were then obscured by a highly scattering layer that uniformly scatters light with a 90-degree divergence (Supplementary Fig. S2 b). Using our method, we successfully reconstructed the 3D structures of these objects (Supplementary Fig. S2 c). These results highlight the robustness of the method in imaging complex 3D continuous objects.

## Discussion

We have demonstrated holographic imaging of spatially incoherent objects through scattering media. Our approach leverages mutually incoherent scattered fields to construct a virtual scattering medium. By numerically propagating the scattered fields through the virtual medium, we effectively compensate for the scattering effects and reconstruct 3D images of the hidden objects. The virtual medium concept is grounded in the well-established physics of forward scattering. It acts as a planar scatterer at an internal plane, capturing the common distortions affecting light from different sources. This suggests that extending the method to volumetric media is feasible, as such common distortions in biological tissue have been experimentally demonstrated[35].

While our method builds on the field-retrieval approach of Baek et al.[32], it provides a framework to recover positional information of sources required for imaging, which was not addressed before. For instance, the partial transformation matrix representation of scattered fields does not encode strict positional information, since the matrix can be constructed in an arbitrary order. Moreover, the previous work could only separate fields of sources confined to the same plane based on intensity correlations, which is insufficient for 3D imaging. Our approach overcomes these limitations by exploiting the field correlations to identify scattered fields from sources distributed across depths, enabling 3D imaging. The virtual scattering medium offers an effective description of the scattering process, thereby circumventing the need for transmission matrix measurement, which is impractical in non-invasive imaging of extended objects. Importantly, our method identifies the effective scattering plane directly from measured wavefronts, requiring no prior knowledge of the medium's geometry. Since the correction is performed fully in the digital domain, it remains robust even under strong scattering, where conventional hardware-based corrections become ineffective.

Our approach is distinguished by its use of inherent field correlations in scattered light[33], enabling non-invasive 3D imaging in regimes inaccessible to speckle intensity correlation methods. Existing methods rely on the scaling of speckle patterns with the axial shift of a point source, which is observed only when the object is positioned millimeters to centimeters from the scatterer[28–31,34,36]. Such requirements are incompatible with microscopy, where scattering occurs over distances of only 10–100 μm. In contrast, our method achieves 3D incoherent imaging through strongly scattering media at these short distances by exploiting field rather than intensity speckle correlations, highlighting its potential for microscopy applications.

The proposed method is a passive imaging approach that does not rely on assumptions about the object, and thus can be extended to

realistic imaging scenarios in fluorescence microscopy. Key considerations for bio-imaging applications include maintaining system stability during measurements and ensuring sufficient signal levels for reliable field retrieval. Several improvements are possible beyond our proof-of-concept demonstration, such as optimizing the setup design to improve transmittance, refining field retrieval algorithms to account for experimental drift[37], and reducing the number of measurements through regularization strategies[38]. Another important factor is keeping the number of effective sources manageable for both data acquisition and computation. This can be addressed by shaping the excitation beam to reduce the number of active fluorescent sources[15,16]. Since the proposed method assumes that light from each source is described by a single field, moderate temporal coherence is required. A narrower bandwidth improves model accuracy but reduces signal strength. When the detection bandwidth exceeds the spectral correlation range of the scattering medium[39], each source contains multiple spectral components, reducing model accuracy. Thus, optimizing the bandwidth based on the properties of the scattering medium will be critical for practical applications. Together, these measures will enable practical implementation in fluorescence microscopy.

In conclusion, our work paves the way for 3D imaging in highly scattering environments. We anticipate that this method will open up new possibilities for imaging and sensing applications involving the scattering of spatially incoherent light.

## Methods

### Field retrieval

Light from independent sources is mutually incoherent. Thus, the total scattered light can be expressed as a set of incoherent fields, $E_n$, where $n$ is the field index. The camera can only measure the sum of intensity, $I = \sum_n |E_n|^2$. To recover the field under this condition, we adopt the mixed-state phase retrieval method introduced in our earlier work[32]. We apply spatial modulation using an SLM placed at the Fourier plane of the fields. This produces the modulated measurement,

$$I_m = \sum_n \left| \mathcal{F}^{-1}\left[ \widetilde{E}_n K_m \right] \right|^2, \tag{5}$$

where $\mathcal{F}$ is the Fourier transform, $\widetilde{E}_n$ is the field at the SLM plane, and $K_m$ is the $m$th random phase pattern displayed on the SLM. Writing the modulated field at the camera plane as $\psi_n^{(m)} = \mathcal{F}^{-1}[\widetilde{E}_n K_m]$, the update rule becomes

$$\psi_n^{(m)} \to \sqrt{\frac{I_m}{\sum_n |\psi_n^{(m)}|^2}} \psi_n^{(m)}, \tag{6}$$

which updates the fields as $\widetilde{E}_n = \mathcal{F}[\psi_n^{(m)}]K_m^*$. This procedure is applied for all modulation patterns and repeated until convergence, as detailed in ref. 32.

### Demixing fields for individual sources

The field retrieval generally yields a linear combination of the scattered fields from different sources rather than the individual fields. This is because intensity measurements cannot uniquely distinguish mutually incoherent contributions. Specifically, the measured intensity of incoherent fields is invariant under any unitary transformation of the fields, $I = \sum_n |E_n|^2 = \sum_m |\sum_n U_{mn} E_n|^2$, where $E_n$ is the field from the $n$th source, $U_{nm}$ is a unitary matrix, and $m$ indices the transformed modes.

To resolve this, we introduce a demixing step that recovers individual fields from such mixtures. This approach exploits the difference in correlation between mixed and demixed fields. As described in Eq. (3), the relative phase between fields from individual sources at the correlation plane can be approximated by a quadratic phase, $\mathcal{E}_n^*(\mathbf{r})\mathcal{E}_m(\mathbf{r}) \approx e^{i\Phi_{nm}(\mathbf{r})}$. This indicates that the relative phase of the mixed fields is expressed by multiple quadratic phase terms. In contrast, the

demixed fields have a relative phase with a single quadratic phase. Using this property, an appropriate unitary matrix $U$ for demixing can be identified. For example, in the case of 2D objects, the relative phase $\Phi_{nm}(\mathbf{r})$ reduces to a linear term. Based on this observation, we define a demixing metric as:

$$M(U) = \sum_{n \neq m} \max_{\mathbf{r} \neq 0} |\mathcal{F}\left[\mathcal{E}_n'^* \mathcal{E}_m'\right](\mathbf{r})|, \tag{7}$$

where $\mathcal{E}_n'(\mathbf{r}) = \sum_{m=1}^N U_{nm} \mathcal{E}_m^{\text{ret}}(\mathbf{r})$, and $\mathcal{E}_n^{\text{ret}}(\mathbf{r})$ is the retrieved field at the correlation plane, defined as $\mathcal{E}_n^{\text{ret}}(\mathbf{r}) = \mathcal{P}_{\hat{z}}\left[E_n^{\text{ret}}(\mathbf{r})\right]$. This metric is maximized when the relative phase of the unitary-transformed fields $\mathcal{E}'(\mathbf{r})$ exhibits a single linear term. We determine the unitary matrix $U$ that maximizes $M$ using the Riemannian optimization algorithm[40]. The resultant unitary matrix effectively demixes the fields, $\mathcal{E}_n(\mathbf{r}) = \sum_{m=1}^N U_{nm} \mathcal{E}_n^{\prime \text{ret}}(\mathbf{r})$ (see Supplementary Fig. S3).

For 3D objects, the demixing metric for the 2D case can be extended to accommodate the quadratic phase of $\Phi_{nm}(\mathbf{r})$. However, rather than directly incorporating the quadratic phase, we employed a simplified approach that leverages local field correlations. By examining field correlations within small areas at the correlation plane, the quadratic phase of $\Phi_{nm}(\mathbf{r})$ can be locally approximated as a linear phase, making demixing possible, similar to the 2D case. Supplementary Fig. S4 illustrates the differences in field correlation (Eq. (8)) using the entire field of view and smaller regions. Locally demixed fields are used to construct a virtual layer at their corresponding regions, and the entire virtual layer is formed by stitching these local constructions. For example, the result in Fig. 4 was obtained by dividing the field of view into a 3 × 3 grid of partially overlapping regions.

### Locating the correlation plane

To determine the position of the internal correlation plane, we propagate the retrieved fields, $E_n^{\text{ret}}$, and evaluate their tilt-correlation at different depths. To achieve this, we used the correlation function defined as:

$$\Gamma(\mathbf{r}, z) = \sum_{n, m} \left| \mathcal{F}\left\{ \mathcal{P}_z[E_n^{\text{ret}}]^* \mathcal{P}_z[E_m^{\text{ret}}] \right\}(\mathbf{r}) \right|^2. \tag{8}$$

Then the axial position of the correlation plane, $\hat{z}$, is identified as the plane with the maximum correlation value:

$$\hat{z} = \arg\max_z \left[ \max_{r \neq 0} \Gamma(\mathbf{r}, z) \right], \tag{9}$$

where the contribution of autocorrelation is eliminated by excluding $\mathbf{r} = 0$ (see Supplementary Fig. S5).

### Experimental setup

The experimental setup is illustrated in Supplementary Fig. S6. For the results shown in Figs. 2 and 3, a laser (Compass 215M-50, Coherent) was used to excite fluorescent beads (1 μm, F8820, Invitrogen), which were immersed in glycerol for the experiments in Fig. 2 and in UV glue (NOA 68, Norland) for those in Fig. 3. The emitted fluorescence was scattered by a 220-grit ground glass diffuser and collected by an objective lens (MPlan N 50 × 0.75, Olympus). Scattered fluorescence was modulated by a spatial light modulator (SLM, X10468-04, Hamamatsu) and recorded by a sCMOS camera (PCO.edge 5.5, PCO) located at the Fourier plane of the SLM. To measure the emission signal while blocking the excitation beam, fluorescence filters (NF533-17 and MF530-43, Thorlabs) were placed in front of the sCMOS camera, with an additional filter (BLP01-532R-25, Semrock) positioned before the SLM. Ground truth images were obtained by directly imaging the fluorescent objects from the opposite side without the scattering media, using a different objective lens (Plan N 20 × 0.4, Olympus) and a

camera (acA5472-17um, Basler) with a fluorescence filter (NF533-17, Thorlabs). For the results in Fig. 4, a digital micromirror device (AJD-4500, Ajile) was installed in the laser path (dashed box in Supplementary Fig. S6), and the axial position of the tube lens (L1 in Supplementary Fig. S6) was adjusted to project point sources at different depths. A bandpass filter (FL532-3, Thorlabs) was placed in front of the sCMOS camera, replacing the previously mentioned filters.

## Data availability

The data supporting this study are available in the Zenodo repository at https://doi.org/10.5281/zenodo.17456705.

## Code availability

Example codes and data are available at https://github.com/comediaLKB/incoherent_holographyor in the Zenodo repository at https://doi.org/10.5281/zenodo.17457640.

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

## Acknowledgements

This research was supported by the European Union's Horizon Europe research and innovation programme MSCA-IF N˚101105899 (Y.B.), H2020 Future and Emerging Technologies 863203 (H.B. and S.G.), and

grant 2024-337799 from the Chan Zuckerberg Initiative DAF, an advised fund of Silicon Valley Foundation (H.B. and S.G.). S.G. is a member of the Institut Universitaire de France.

## Author contributions

Y.B. conceived the project, performed the experiment, and analyzed the data. Y.B., H.B., and S.G. designed the experiment, interpreted the results and wrote the manuscript.

## Competing interests

The authors declare no competing interests.
