## [Transparent Peer Review file · Nature Communications]

Three-dimensional holographic imaging of incoherent objects through scattering media

Corresponding Author: Dr YoonSeok Baek

Version 0:

Reviewer comments:

Reviewer #1

(Remarks to the Author)

In this manuscript, the authors propose a holographic imaging technique that enables three-dimensional (3D) reconstruction of spatially incoherent objects hidden behind scattering media. The approach relies on retrieving multiple mutually incoherent scattered fields using spatial modulation of light and a phase retrieval algorithm. By analyzing the correlation between these scattered fields, a “virtual scattering medium” is numerically constructed, replicating the phase distortions imposed by the actual scattering layer. The retrieved incoherent fields are then numerically propagated through this virtual medium to compensate for the scattering effects, thereby enabling accurate 3D image reconstruction of the hidden object. The method is validated experimentally using both fluorescent beads and synthetic 3D incoherent objects, demonstrating its robustness and non-invasive nature for volumetric imaging in turbid environments. While the manuscript presents a technically sound approach, several suggestions and comments need to be addressed in the revised version.

Comment 1: The current manuscript heavily builds upon the authors' earlier paper (Nature Photonics, 17, 1114 (2023)) where mutually incoherent fields were retrieved and used for phase conjugation and energy delivery. While this paper extends the concept to 3D reconstruction, the underlying experimental strategy, including field retrieval and demixing, is largely shared. The authors must explicitly highlight what conceptual and methodological advances go beyond the previous work.

Comment 2: The introduction of a virtual scattering medium is the central novelty here, yet its mathematical formulation and physical interpretation are not rigorously established. The manuscript would benefit from a deeper theoretical explanation of how this virtual medium generalizes or improves over conventional scattering correction techniques.

Comment 3: There is no quantitative evaluation of reconstruction quality—no PSNR, SSIM, resolution, or depth accuracy analysis is presented. Such metrics are essential to validate the effectiveness and compare it with existing methods.

Comment 4: The manuscript lacks benchmarking against other recent 3D imaging methods through scattering media. A comparative study would strengthen the significance of this work.

Comment 5: Although the method is described as useful for microscopy or biological imaging, the authors have not shown any results using real biological samples. Including either a simulated example or a discussion on how this technique would work with real biological tissue would help demonstrate its practical value.

Comment 6: The method seems to use something similar to a partial transmission matrix, like in the authors' earlier work (Nature Photonics, 17, 1114 (2023)). It would be helpful if the authors clearly explain how their “virtual medium” is different from or related to other imaging methods that use the transmission matrix.

Comment 7: In the caption of Figure 3, part (c) is mentioned, but no corresponding subfigure is presented in the figure itself. Additionally, the current labeling of subfigures is inconsistent. The subfigures labeled as (b) and (c) should be corrected to (d) and (e), and those currently labeled as (d) and (e) should be relabeled as (f) and (g) respectively.

(Remarks on code availability)

highly required

Reviewer #2

(Remarks to the Author)

The authors image fluorescent beads through a ground glass diffuser. The method consists of a series of highly advanced steps for measuring the fields without reference, demixing the fields from different emitters, reconstructing the scattering layer

from the measured fields, and back-propagating the fields to reconstruct an image of the beads.

I think this is definitely very nice work. However, some improvements are needed before acceptance.

- 1) The source code is not documented, making it very hard to know how to use it. Also the format of the input data ('E_ret') is not specified. It seems that code is missing to come from the set of DMD patterns and corresponding camera images (i.e. the raw data) to the 'E_ret' data. Also, the functions 'prop_ys' and 'imagescX' were not defined in my Matlab version, so I could not run the code.
- 2) The ptychography-like method to measure optical fields without external reference was published in ref 32. Still, there also should be an explanation of the method in the current manuscript to make it readable as a stand-alone paper.
- 3) The authors assume that the fields E_n and E_m for different sources can be measured individually. If this is the case, why do they need to be demixed? i.e. why is the matrix U_{nm} needed, and what do the indices mean? (they cannot both be sources). This part of the method definitely needs a more thorough explanation, and it should be made clear what is new compared to [32].
- 4) The concept of an 'incoherent object' is not clearly defined. When reading more carefully, it is a small set of point emitters, filtered with a very narrow band filter combination. Under these circumstances, indeed, a speckle can be seen. In general, however, an object that emits incoherent light will not produce a speckle, and the term 'incoherent speckle' is a contradiction in terminis. The authors should discuss the limitations of their method, considering both spatial and temporal coherence.
- 5) The authors imply that their method extends beyond the optical memory effect (e.g. by saying the sources are so close to the scattering layer that there is no convolution with a single speckle pattern). However, by approximating the scattering layer by a single mathematical plane, the authors make exactly this assumption, which appears to work just fine, meaning that this work relies fully on the optical memory effect.

Various:

- 'if the axial extent of the incoherent object is not significantly larger than the scattering mean free path...'. This statement is not immediately trivial, and should be supported by a mathematical derivation or a citation.
- I don't understand why the experiment with the synthetic object is needed. The real experiment (figure 3) is convincing enough, so why add 'fake' data?
- Figure 3b is mislabeled/missing.
- Some minor typos (plural/singular)

(Remarks on code availability)

The code was is not documented, seemed incomplete (input was not raw data but already preprocessed), and I could not execute it due to missing functions.

Version 1:

Reviewer comments:

Reviewer #1

(Remarks to the Author)

(Remarks on code availability)

REVIEWER COMMENTS

Reviewer #1 (Remarks to the Author):

In this manuscript, the authors propose a holographic imaging technique that enables three-dimensional (3D) reconstruction of spatially incoherent objects hidden behind scattering media. The approach relies on retrieving multiple mutually incoherent scattered fields using spatial modulation of light and a phase retrieval algorithm. By analyzing the correlation between these scattered fields, a “virtual scattering medium” is numerically constructed, replicating the phase distortions imposed by the actual scattering layer. The retrieved incoherent fields are then numerically propagated through this virtual medium to compensate for the scattering effects, thereby enabling accurate 3D image reconstruction of the hidden object. The method is validated experimentally using both fluorescent beads and synthetic 3D incoherent objects, demonstrating its robustness and non-invasive nature for volumetric imaging in turbid environments. While the manuscript presents a technically sound approach, several suggestions and comments need to be addressed in the revised version.

We appreciate the reviewer for the thorough evaluation and valuable feedback. We have addressed comments and incorporated the suggested improvements.

Comment 1: The current manuscript heavily builds upon the authors’ earlier paper (Nature Photonics, 17, 1114 (2023)) where mutually incoherent fields were retrieved and used for phase conjugation and energy delivery. While this paper extends the concept to 3D reconstruction, the underlying experimental strategy, including field retrieval and demixing, is largely shared. The authors must explicitly highlight what conceptual and methodological advances go beyond the previous work.

We thank the reviewer for the insightful comments. As noted, the manuscript builds upon the field retrieval methods introduced in our previous work. The earlier study focused on retrieving scattered fields of the hidden sources, while the exact spatial distribution of those sources remained unknown. This information was sufficient to focus light on source positions, but not for reconstructing images. In contrast, the current work goes well beyond by demonstrating that the retrieved information can be exploited to digitally reconstruct hidden objects. This is achieved through the use of wavefront correlations and the introduction of a virtual scattering medium. While the previous work included a demixing step based on 2D intensity correlations, that approach was limited to sources lying on the same plane and could not be extended to 3D objects. In this work, the demixing is performed using field correlations, which enables 3D imaging. We have clarified this distinction in the discussion section as follows:

While our method builds on the field-retrieval approach of [32], it provides a framework to recover positional information of sources required for imaging, which was not addressed before. For instance, the partial transformation matrix representation of scattered fields does not encode strict positional information, since the matrix can be constructed in an arbitrary order. Moreover, the previous work could only separate fields of sources confined to the same plane based on intensity correlations, which is insufficient for 3D imaging. Our approach overcomes these limitations by exploiting the field correlations to identify scattered fields from sources distributed across depths, enabling 3D imaging. (...)

Comment 2: The introduction of a virtual scattering medium is the central novelty here, yet its mathematical formulation and physical interpretation are not rigorously established. The manuscript would benefit from a deeper theoretical explanation of how this virtual medium generalizes or improves over conventional scattering correction techniques.

We thank the reviewer bringing up this important point. The virtual scattering medium introduced in this work can be understood as a wavefront distortion located at a specific plane. This can be made explicitly using the field correlation in forward scattering media in Ref. [33]. Equation (2) shows that scattered fields from two neighboring sources are maximally correlated at a specific plane after adjusting wavefront tilt:

$$\max|C| = \left| \int \mathcal{E}_n(r) \mathcal{E}_m^*(r) e^{i\Delta k_{opt} \cdot r} d^2r \right|.$$

This allows one field to be expressed in terms of the other:

$$\mathcal{E}_n(r) \propto \mathcal{E}_m(r) e^{-i\Delta k_{opt} \cdot r} + M_n,$$

Where M_n is a small term uncorrelated with \mathcal{E}_m . The scattered field can be written as the product of the field in the absence of scattering, $h_n(r)$, and a distortion, $S_n(r)$, such that $\mathcal{E}_n(r) = h_n(r)S_n(r)$. Substituting into the relation above gives:

$$h_n(r)S_n(r) \propto h_m(r) e^{-i\Delta k_{opt} \cdot r} S_m(r) + M_n$$

Using $h_n(r) = h_m(r) e^{-i\Delta k_{opt} \cdot r}$ and dividing by h_n , we obtain

$$S_n(r) \propto S_m(r) + \frac{M_n}{h_n}.$$

This shows the existence of a shared distortion in all wavefronts. We can express the average distortion across all sources as

$$S_o(r) = \sum_n S_n(r).$$

When a scattered field is backpropagated through S_o , the scattering-free wavefront is recovered:

$$\mathcal{E}_n^*(r) S_o(r) = |S_o(r)|^2 h_n^*(r) + M',$$

where M' is a noise term arising from the uncorrelated terms. Propagating this to the source plane produces a sharp focus at the source location with background noise. This result shows that the object image can be reconstructed by backpropagating all the wavefronts through a virtual medium that reflects the effect of S_o . We have included this explanation in the Supplementary section ‘‘Shared distortion in scattered fields’’.

The key advantage of our approach is that the effective distortion plane is identified *a posteriori* from wavefront correlations, without requiring prior knowledge of the scattering geometry. In many realistic scenarios this plane is otherwise difficult to determine. Moreover, the complex distortion is extracted entirely digitally rather than by experimentally determining the wavefront correction, which allows our method to handle stronger scattering more effectively than other experimental approaches. Existing computational imaging methods rely on speckle intensity correlations, which can reveal the relative positions of sources under specific conditions. Our approach, in contrast, exploits wavefront correlations and remains effective even when the intensity correlations do not reveal clear object information, as shown in the subsets of Fig. 2a and 2b. This makes it broadly applicable to imaging over larger areas, under stronger scattering, and in three dimensions. To clarify the strength of our method, we have added an explanation in the discussion section:

(...) The virtual scattering medium offers an effective description of the scattering process, thereby circumventing the need for transmission matrix measurement, which is impractical in non-invasive imaging of extended objects. Importantly, our method identifies the effective scattering plane directly from measured wavefronts, requiring no prior knowledge of the medium's geometry. Since the correction is performed fully in the digital domain, it remains robust even under strong scattering where conventional hardware-based corrections become ineffective.

Comment 3: There is no quantitative evaluation of reconstruction quality—no PSNR, SSIM, resolution, or depth accuracy analysis is presented. Such metrics are essential to validate the effectiveness and compare it with existing methods.

We thank the reviewer for the suggestion to include a quantitative evaluation. Our work addresses the challenging problem of 3D incoherent imaging through highly scattering media on a microscopic scale, which has been elusive so far. While the reconstructed images already show clear agreement with the ground truth, we have included PSNR and SSIM analyses for completeness. These metrics require matching images with different bit depth and pixel size, and their reliability is further limited by the sparsity of the images, differences in resolution, and the position-dependent transmittance of scattering media. We therefore provide these values only as reference, as they are not universal indicators of reconstruction quality in this context. The PSNR and SSIM values are 35.5 dB and 0.96 for Fig. 2 e, g, 38.6 dB and 0.93 for Fig. 2f, h, 40.2 dB and 0.91 for Fig. 3c, e, and 40.1 dB and 0.95 for Fig. 3d, f. These results have been included in the text describing for Fig. 2 and Fig. 3.

The resolution in our method is determined by the spatial frequency content of the retrieved field, which is limited by the resolution of the optical system or its numerical aperture (NA). Since our reconstruction is based on direct backpropagation of light, it does not artificially generate spurious image details. In our experiment, the effective detection NA was 0.35, corresponding to a resolution of $0.96\ \mu\text{m}$. The source in Fig. 4c has a size (FWHM) of $1.04\ \mu\text{m}$, with good agreement with the expected resolution. For depth and positional accuracy, we directly compared the reconstructed source positions with their expected locations using the data in Fig. 4, marking the exact positions with red + symbols overlaid on the MIP images:

The comparison accounts for the angular rotation due to the viewing angle (7-degree rotation relative to the x-axis) and shows that the positional accuracy is within the diffraction limit range. We added an explanation of the resolution and positional accuracy in the main text describing Fig. 4:

Our reconstruction is analogous to digital holography once the virtual scattering medium is identified. The resolution is set by the spatial frequency content of the light, not by the reconstruction method itself, since our approach numerically propagates the

retrieved fields. For our experimental setup with an effective NA of 0.35, the corresponding resolution is 0.96 μm . The full width at half maximum of a source in Fig. 4c is 1.04 μm , showing close agreement. To demonstrate positional accuracy, we overlaid the expected positions of the point sources on the MIP images (Fig. S1). To account for the difference in viewing angle between illumination and detection setup, we applied a rotation correction for optimal alignment. The positional error in all directions is comparable to or below the diffraction limit.

The comparative figure is included as Fig. S1 in the revised manuscript.

Comment 4: The manuscript lacks benchmarking against other recent 3D imaging methods through scattering media. A comparative study would strengthen the significance of this work.

We thank the reviewer for the comment on benchmarking. Comment 4 is closely related to Comment 3 regarding quantitative comparisons with other 3D imaging approaches. As explained in the manuscript, existing methods for 3D incoherent imaging through scattering media, such as speckle correlation techniques, require the object to be located far from the scatterer, typically in the millimeter to centimeter range. These conditions are incompatible with microscopy, where scattering occurs over much shorter distances of 10–100 μm . Our work demonstrates 3D incoherent imaging through highly scattering media at these short distances, highlighting its relevance for microscopy applications. Since existing methods are not applicable in this regime, a direct benchmarking is not feasible. To address the reviewer's point, we have further clarified this in the Discussion section:

Our approach is distinguished by its use of inherent field correlations in scattered light, enabling non-invasive 3D imaging in regimes inaccessible to speckle-intensity-correlation methods. Existing methods rely on the scaling of speckle patterns with the axial shift of a point source, which is observed only when the object is positioned millimeters to centimeters from the scatterer. Such requirements are incompatible with microscopy, where scattering occurs over distances of only 10–100 μm . In contrast, our method achieves 3D incoherent imaging through strongly scattering media at these short distances by exploiting field rather than intensity speckle correlations, highlighting its potential for microscopy applications.

Comment 5: Although the method is described as useful for microscopy or biological imaging, the authors have not shown any results using real biological samples. Including either a simulated example or a discussion on how this technique would work with real biological tissue would help demonstrate its practical value.

We thank the reviewer for the valuable feedback. Our method is a passive imaging approach that makes no assumptions about the object. The results in the main text (Fig. 4) and Supplementary section (Fig. S2) already demonstrate the ability to image arbitrary source distributions without prior knowledge. An additional simulation with different samples would merely redistribute sources and would not provide new insight for this proof-of-concept work. For this reason, we considered it more meaningful to expand the discussion to clarify the applicability of the method to biological imaging and to outline the practical adaptations required for experimental implementation. These include ensuring system stability, handling sensitive specimens, optimizing imaging conditions, implementing algorithmic improvements to reduce measurements, and adopting optimized excitation strategies. In the revised discussion we have added:

The proposed method is a passive imaging approach that does not rely on assumptions about the object, and thus can be extended to realistic imaging scenarios in fluorescence microscopy. Key considerations for bio-imaging applications include maintaining system stability during measurements and ensuring sufficient signal levels for reliable field retrieval. Several improvements are possible beyond our proof-of-concept demonstration, such as optimizing setup design to improve transmittance, refining field retrieval algorithms to account for experimental drift, and reducing the number of measurements through regularization strategies. Another important factor is keeping the number of effective sources manageable for both data acquisition and computation. This can be addressed by shaping the excitation beam to reduce the number of active fluorescent sources. Since the proposed method assumes that light from each source is described by a single field, moderate temporal coherence is required. A narrower bandwidth improves model accuracy but reduces signal strength. When the detection bandwidth exceeds the spectral correlation range of the scattering medium [39], each source contains multiple spectral components, reducing model accuracy. Thus, optimizing the bandwidth based on the properties of the scattering medium will be critical for practical applications. Together, these measures will enable practical implementation in fluorescence microscopy.

Comment 6: The method seems to use something similar to a partial transmission matrix, like in the authors' earlier work (Nature Photonics,17, 1114 (2023)). It would be helpful if the authors clearly explain how their "virtual medium" is different from or related to other imaging methods that use the transmission matrix.

We thank the reviewer for the insightful comment. In our previous work, the partial transmission matrix was constructed as a matrix representation of the retrieved scattered fields, with the vectorized fields forming the rows or columns. Since the present work also retrieves scattered fields for imaging, the two approaches are equivalent in terms of information content, as both start from the same measured data. Importantly, however, the partial transmission matrix does not directly encode the image of the object. It contains an intrinsic ambiguity, since different orderings or different unitary combination of the vectorized fields lead to the same focusing results. While this ambiguity does not affect light delivery, it means that the positional information of the sources is not explicitly resolved. Our method addresses this limitation by exploiting correlations in the scattered fields as a virtual scattering layer. This provides a complete description that includes positional information through modeling the full system with the virtual scatterer and propagation operators.

Regarding other imaging methods based on the transmission matrix, these approaches require calibration of the full transmission matrix prior to imaging. This calibration involves generating a complete set of coherent light bases on one plane and measuring the corresponding output fields, which is impractical in many scenarios where arbitrary wavefront generation inside or through scattering media is not feasible. For example, the number of measurements in our scheme scales with the number of sources, while other methods typically scale with the number of modes of the scattered fields, which is generally leads to much poorer scaling as the wavefront complexity increases. Moreover, such methods typically depend on coherent laser illumination and interferometric measurements, making extension to incoherent imaging highly nontrivial. In contrast, our work achieves a similar capability without interferometric calibration, by directly retrieving and analyzing the scattered fields of hidden incoherent sources. This substantially extends the scope of transmission-matrix-based imaging.

We have clarified these distinctions and their implications in the discussion section of the revised manuscript.

While our method builds on the field-retrieval approach of [32], it provides a framework to recover positional information of sources required for imaging, which was not addressed before. For instance, the partial transformation matrix representation of scattered fields does not encode strict positional information, since the matrix can be constructed in an arbitrary order. Moreover, the previous work could only separate fields of sources confined to the same plane based on intensity correlations, which is insufficient for 3D imaging. Our approach overcomes these limitations by exploiting the field correlations to identify scattered fields from sources distributed across depths, enabling 3D imaging. The virtual scattering medium offers an effective description of the scattering process, thereby circumventing the need for transmission matrix calibration, which is impractical in non-invasive imaging. (...)

Comment 7: In the caption of Figure 3, part (c) is mentioned, but no corresponding subfigure is presented in the figure itself. Additionally, the current labeling of subfigures is inconsistent. The subfigures labeled as (b) and (c) should be corrected to (d) and (e), and those currently labeled as (d) and (e) should be relabeled as (f) and (g) respectively.

We thank the reviewer for the careful review. We have corrected the labels of Figure 3.

Reviewer #1 (Remarks on code availability):

highly required

We have updated the repository to provide documented codes along with raw input data and all necessary functions.

Reviewer #2 (Remarks to the Author):

The authors image fluorescent beads through a ground glass diffuser. The method consists of a series of highly advanced steps for measuring the fields without reference, demixing the fields from different emitters, reconstructing the scattering layer from the measured fields, and back-propagating the fields to reconstruct an image of the beads. I think this is definitely very nice work. However, some improvements are needed before acceptance.

We thank the reviewer for the positive feedback on our work. We have addressed each comment and incorporated the suggested improvements as detailed below.

Comment 1: The source code is not documented, making it very hard to know how to use it. Also the format of the input data ('E_ret') is not specified. It seems that code is missing to come from the set of DMD patterns and corresponding camera images (i.e. the raw data) to the 'E_ret' data. Also, the functions 'prop_ys' and 'imagescX' were not defined in my Matlab version, so I could not run the code.

Thank you for the careful review of the code and for pointing out the missing parts. We've updated the repository with proper documentation, included the missing functions, and added

a separate script (phase_retrieval_example.m) for retrieving the field (E_{ret}) from the raw intensity images (image_data.mat).

Comment 2: The ptychography-like method to measure optical fields without external reference was published in ref 32. Still, there also should be an explanation of the method in the current manuscript to make it readable as a stand-alone paper.

We thank the reviewer for the suggestion. To make the paper more standalone, we've included an explanation of the field retrieval process in the Methods section:

In the main text, we added:

(...) Then the resulting speckle pattern is analyzed through an iterative algorithm to retrieve mutually incoherent fields of independent incoherent sources (see Methods for details on the field retrieval and analysis).

In the Method section, under 'Field retrieval', we added:

Light from independent sources is mutually incoherent. Thus, the total scattered light can be expressed as a set of incoherent fields, E_n , where n is the field index. The camera can only measure the sum of intensity, $I = \sum_n |E_n|^2$. To recover the field under this condition, we adopt the mixed-state phase retrieval method introduced in our earlier work [32]. We apply spatial modulation using an SLM placed at the Fourier plane of the fields. This produces the modulated measurement,

$$I_m = \sum_n |\mathcal{F}^{-1}[\widetilde{E}_n K_m]|^2,$$

where \mathcal{F} is Fourier transform, \widetilde{E}_n is the field at the SLM plane, and K_m is the m th random phase pattern displayed on the SLM. Writing the modulated field at the camera plane as $\psi_n^{(m)} = \mathcal{F}^{-1}[\widetilde{E}_n K_m]$, the update rule becomes

$$\psi_n^{(m)} \rightarrow \sqrt{\frac{I_m}{\sum_n |\psi_n^{(m)}|^2}} \psi_n^{(m)},$$

which updates the fields as $\widetilde{E}_n = \mathcal{F}[\psi_n^{(m)}] K_m^$. This procedure is applied for all modulation patterns and repeated until convergence, as detailed in [32].*

Comment 3: The authors assume that the fields E_n and E_m for different sources can be measured individually. If this is the case, why do they need to be demixed? i.e. why is the matrix U_{nm} needed, and what do the indices mean? (they cannot both be sources). This part of the method definitely needs a more thorough explanation, and it should be made clear what is new compared to [32].

We thank the reviewer for the comment. The demixing step was introduced to address the ambiguity in intensity measurement. For multiple incoherent sources, the measured intensity is the sum of individual contributions

$$I = \sum_n |E_n|^2,$$

where E_n is the field from source n . Importantly, the intensity remains unchanged under any other unitary transformation of the fields:

$$I = \sum_n |E_n|^2 = \sum_m \left| \sum_n U_{mn} E_n \right|^2,$$

where U_{mn} is a unitary matrix and m is the index of the transformed modes. This shows the fundamental ambiguity in defining which specific set of incoherent modes constitutes the mixture. A well-known example of this is depolarized light, which can equivalently be described as a mixture of horizontal and vertical polarizations or as a mixture of left and right circular polarizations. Similarly, in our case, the total light can be represented as a mixture of scattered fields from independent sources or as a mixture of their unitary transformations. As a result, field retrieval algorithm typically recovers a unitary mixture rather than the fields from individual sources. To resolve this, we introduced a demixing step to recover the fields associated with each source. We clarified the necessity of the demixing by expanding the explanation in the relevant Method section:

The field retrieval generally yields a linear combination of the scattered fields from different sources rather than the individual fields. This is because intensity measurements cannot uniquely distinguish mutually incoherent contributions. Specifically, the measured intensity of incoherent fields is invariant under any unitary transformation of the fields, $I = \sum_n |E_n|^2 = \sum_m |\sum_n U_{mn} E_n|^2$, where U_{nm} is a unitary matrix and m indexes the transformed modes.

To resolve this, we introduce a demixing step that recovers individual fields from such mixtures. (...)

Regarding the comparison with [32], our previous work did not address imaging through scattering media. Although [32] includes a demixing step, that previous approach relies on intensity correlations and is limited to sources are confined to a single 2D plane. In contrast, our method uses the field-correlation, which enables demixing even when sources are distributed across multiple depths and a broader spatial extent, thereby making 3D imaging possible. In the discussion section, we have clarified the distinction between our method and that of [32]:

While our method builds on the field-retrieval approach of [32], it provides a framework to recover positional information of sources required for imaging, which was not addressed before. For instance, the partial transformation matrix representation of scattered fields does not encode strict positional information, since the matrix can be constructed in an arbitrary order. Moreover, the previous work could only separate fields of sources confined to the same plane based on intensity correlations, which is insufficient for 3D imaging. Our approach overcomes these limitations by exploiting the field correlations to identify scattered fields from sources distributed across depths, enabling 3D imaging.

Comment 4: The concept of an 'incoherent object' is not clearly defined. When reading more carefully, it is a small set of point emitters, filtered with a very narrow band filter combination. Under these circumstances, indeed, a speckle can be seen. In general, however, an object that emits incoherent light will not produce a speckle, and the term 'incoherent speckle' is a contradiction in terminis. The authors should discuss the limitations of their method, considering both spatial and temporal coherence.

We thank the reviewer for the comment.

Regarding the term ‘incoherent object’, it refers to an object that reflects or emits incoherent light and has been commonly used in the field of holography [J. Opt. Soc. Am. 55, 1555_1-1556 (1965); Opt. Lett. 37, 2694-2696 (2012)]. In response, we have clarified its meaning in the beginning of the main text:

For imaging objects that reflect or emit spatially incoherent light such as fluorescent sources, referred to as incoherent objects, wavefront shaping techniques (...)

Regarding the comment on bandpass filters, we detected about 10 nm of the fluorescence emission, which corresponds to a coherence length of 30 μm or a coherence time of 100 fs. While this filtering improves the temporal coherence of light, but it does not make light from different sources produce a stable interference pattern over typical imaging integration time (~ 1 second). Each source therefore generates a high-contrast speckle pattern, while the total intensity is simply the sum of those patterns. As a result, the overall contrast is lower than that of a fully coherent light.

Regarding the comment on ‘incoherent speckle’, we agree that spatially fully incoherent light does not produce contrast. However, with finite number (N) of sources, the measured image is the sum of N coherent speckle patterns. This leads to a speckle pattern with reduced contrast that scales with $1/\sqrt{N}$, which we had referred to as “incoherent speckle”. To avoid confusion, we have replaced this term with “low-contrast speckle” in the revised manuscript.

Regarding the comment on spatiotemporal coherence, our method assumes that light originating from each source is represented by a single field, while light from different sources is mutually incoherent. Since the fluorescence emission of different sources is inherently incoherent, only moderate temporal coherence is required to ensure that each scattered field remains spatially coherent. This condition is satisfied in our experiments by detecting light with a 10 nm bandwidth. The manuscript already states this condition by noting that we are imaging and retrieving spatially incoherent light. We have further clarified it and discuss limitations in the Discussion section.

Since the proposed method assumes that light from each source is described by a single field, moderate temporal coherence is required. A narrower bandwidth improves model accuracy but reduces signal strength. When the detection bandwidth exceeds the spectral correlation range of the scattering medium [Optics express 27, 28384 (2019)], each source contains multiple spectral components, reducing model accuracy. Thus, optimizing the bandwidth based on the properties of the scattering medium will be critical for practical applications.

Comment 5: The authors imply that their method extends beyond the optical memory effect (e.g. by saying the sources are so close to the scattering layer that there is no convolution with a single speckle pattern). However, by approximating the scattering layer by a single mathematical plane, the authors make exactly this assumption, which appears to work just fine, meaning that this work relies fully on the optical memory effect.

We thank the reviewer for the insightful comment. Our method is indeed based on the memory effect. The key difference is that we use the field correlation, whereas the existing methods rely on intensity information, which does not exploit the available correlation. We intended to highlight this difference and realized that referring to the existing methods as the “axial memory effect” can be misleading. To clarify, we have replaced “axial memory effect” with “speckle intensity correlation” in the revised manuscript:

“This enables non-invasive 3D imaging in a wider range of scenarios, unlike methods based on speckle intensity correlation, (...)”.

Minor Comments:

- 'if the axial extent of the incoherent object is not significantly larger than the scattering mean free path...'. This statement is not immediately trivial, and should be supported by a mathematical derivation or a citation.

We thank the reviewer for the comment. In our response to Comment 2 of Reviewer #1, we showed mathematically that our method captures the common distortions in the fields from different sources by expressing the scattered fields as the product of the scattering-free wavefront h_n and distortion S_n :

$$\mathcal{E}_n(r) = h_n(r)S_n(r).$$

According to the definition of the scattering mean free path, if the axial extent of the object ($\Delta z = z_{max} - z_{min}$) is small, the light from the deepest source ($z = z_{max}$) does not experience scattering until it reaches the plane of the shallowest source ($z = z_{min}$). In other words, propagation between the two planes can be approximated as free-space. Under this condition, the scattered fields can still be expressed as before, with h_n now corresponds to a spherical wavefront determined by the 3D position of the sources relative to the correlation plane. This conclusion follows directly from the definition of the scattering mean free path. We have clarified this point in the revised main text:

Specifically, if the axial extent of the incoherent object is not significantly larger than the scattering mean free path, scattering within the object's axial size can be neglected. Under this assumption, the fields retain the form $S_o(r)h_n(r)$, with h_n determined by the 3D positions of the sources. (...)

- I don't understand why the experiment with the synthetic object is needed. The real experiment (figure 3) is convincing enough, so why add 'fake' data?

We agree that Fig. 3 provides a convincing proof of concept. The experiment in Fig. 4 was included to demonstrate that the method can handle sources at multiple depths, a scenario that is easier to demonstrate with synthetic data. We appreciate finding the real experiment sufficient, and we hope the additional example provides useful context.

- Figure 3b is mislabeled/missing.

We have corrected the legend and label of Fig. 3 in the revised manuscript.

- Some minor typos (plural/singular)

We have corrected minor typos in the revised manuscript.

Reviewer #2 (Remarks on code availability):

The code was is not documented, seemed incomplete (input was not raw data but already preprocessed), and I could not execute it due to missing functions.

We have updated the repository to provide documented codes along with raw input data and all necessary functions.